# VASA-3D: Lifelike Audio-Driven Gaussian Head Avatars from a Single Image

**Sicheng Xu**[*], **Guojun Chen**[*], **Jiaolong Yang**[†],
**Yu Deng**, **Yizhong Zhang**, **Stephen Lin**, **Baining Guo**
Microsoft Research Asia
{sichengxu,guoch,jiaoyan,yizzhan,dengyu,stevelin,bainguo}@microsoft.com

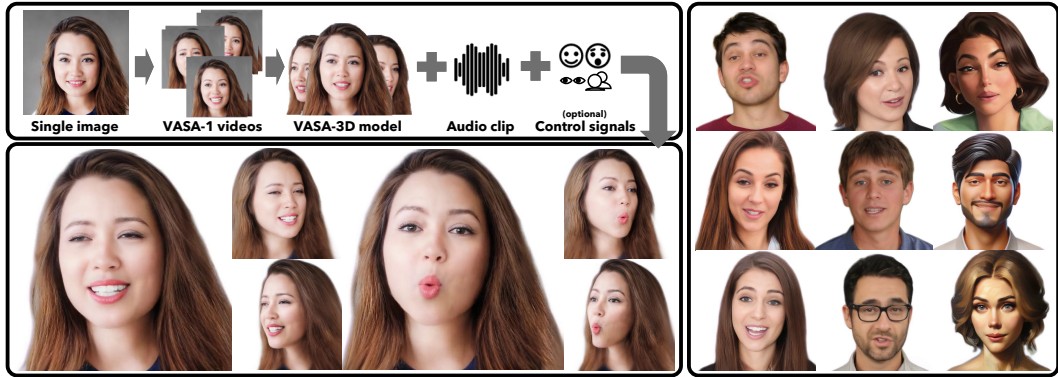

Figure 1: Given a single portrait image, our approach produces an animatable 3D Gaussian head that can be driven with any speech audio clip to generate lifelike, free-viewpoint talking face videos in real time. It adapts the powerful motion latent space of VASA-1 [1] to 3D, and leverages the high realism of VASA-1 in 2D video generation to train the 3D head model. (*Note: all the portrait images in this document are virtual, non-existing identities generated by [2, 3]. See our supplemental material for generated video samples with audio.*)

## Abstract

We propose VASA-3D, an audio-driven, single-shot 3D head avatar generator. This research tackles two major challenges: capturing the subtle expression details present in real human faces, and reconstructing an intricate 3D head avatar from a single portrait image. To accurately model expression details, VASA-3D leverages the motion latent of VASA-1 [1], a method that yields exceptional realism and vividness in 2D talking heads. A critical element of our work is translating this motion latent to 3D, which is accomplished by devising a 3D head model that is conditioned on the motion latent. Customization of this model to a single image is achieved through an optimization framework that employs numerous video frames of the reference head synthesized from the input image. The optimization takes various training losses robust to artifacts and limited pose coverage in the generated training data. Our experiment shows that VASA-3D produces realistic 3D talking heads that cannot be achieved by prior art, and it supports the online generation of 512×512 free-viewpoint videos at up to 75 FPS, facilitating more immersive engagements with lifelike 3D avatars.

---

[*]Equal contributions. [†] Corresponding author.

39th Conference on Neural Information Processing Systems (NeurIPS 2025).

# 1 Introduction

Advances in generating 3D head avatars are revolutionizing digital interaction, effectively bridging the gap between physical presence and virtual engagement. Vivid representations of human faces serve to enhance various applications, ranging from virtual reality and gaming to remote education and online meetings. By conveying realistic facial expressions and movements, 3D head avatars can foster a deeper sense of connection within virtual environments, making interactions more personal and engaging, and significantly improving user experience and immersion.

Most recent research on 3D head avatars utilize parametric head representations derived from 3D scans [4, 5, 6, 7, 8, 9, 10, 11, 12, 13, 14, 15, 16, 17, 18, 19, 20]. The model's shape and motion are personalized to image or video data of a reference face, and then the avatar animation is driven by an audio or video track of what the avatar will say. Despite the impressive developments to date, significant challenges still remain. One major issue with current methods is that their output often lacks the nuanced motion and subtle expressions of real human faces, resulting in less visually compelling facial dynamics. Additionally, a vast majority of existing methods [4, 21, 5, 6, 7, 10, 11, 12, 13, 22, 16, 17, 23, 18] require video or multiview data of the reference face for avatar modeling, limiting their utility.

In this paper, we present VASA-3D, an audio-driven 3D head avatar generator that transforms a single portrait image into a lifelike 3D talking head, synchronized with any speech audio input. The head avatar is modeled with 3D Gaussian splatting [24, 17], which ensures multiview consistency and facilitates real-time audio-driven animation and free-view rendering. Notably, the model captures and conveys dynamic expression details with a degree of realism that markedly exceeds current state-of-the-art techniques.

We observe that the expression terms of parametric head representations used for 3D head avatars, such as 3DMM [25, 26] and FLAME [27], are modeled on 3D scans of just a few hundred subjects. To model more diverse and detailed facial dynamics at minimal acquisition cost, VASA-3D instead takes advantage of 2D head videos, which are abundant online. Specifically, it employs the motion latent of VASA-1 [1], which has been trained on data from 9.5K subjects, to capture rich facial dynamics. This motion latent, though learned on 2D data, encodes implicit 3D structure, and a key contribution of our work is in translating it to a 3D avatar. We accomplish this by first mapping it to the parameters of a FLAME head model, on which 3D Gaussians are bound as done in [17]. Although FLAME's expression parameters are modeled on just hundreds of 4D face captures, VASA-3D addresses this limitation by next predicting dense, freeform Gaussian deformations that are conditioned on the motion latent, thereby enabling the generation of more expressive 3D dynamic heads.

Our latent motion controlled Gaussian avatar offers great potential for 3D talking head synthesis, but customizing it to just a single portrait image poses a challenging problem. Existing methods that personalize a head avatar representation using a single image [8, 14, 15, 19, 20, 28] encode facial expression via a parametric head model, thus limiting expressiveness. Our solution is to utilize a pretrained portrait video generation model, namely VASA-1 [1], to transform the reference image into a collection of frames with varied facial expressions and head poses, and then fit the avatar to these frames. As there exist limitations with this synthetic data, as well as overfitting issues due to dense deformation, we have developed an optimization framework with various training losses designed to robustly train the model despite these problems.

VASA-3D represents a significant step forward in 3D head avatar synthesis, capitalizing on 2D head videos to enrich its model of facial dynamics and allowing customization of this advanced model using only a single portrait image. The effectiveness and realism of VASA-3D is validated through various experiments, where it demonstrates clear superiority over recent techniques. By creating lifelike avatars that accurately reflect human expressions and facial motions, this approach paves the way for more immersive and engaging virtual experiences.

# 2 Related Work

**3D Face and Head Representations.** A common representation for 3D head avatars is parametric mesh-based models such as 3DMM [25] and FLAME [27]. These models are built on a collection of 3D head scans, from which the principal components of shape with respect to identity and expression are used as bases for modeling head and face geometry. Though compact and efficient, these

parametric models provide low-fidelity mesh representations and limited detail for facial expressions, whose bases are derived from scanned data of only hundreds of subjects.

An alternative approach based on neural radiance fields (NeRFs) [29] does not explicitly represent geometry but instead stores the radiance field of a head in a neural network. With these radiance values, head appearance at novel views can be synthesized by volumetric rendering. This approach has led to many works that can generate highly photorealistic head avatars [4, 9, 7, 13]. However, NeRF-based methods often require multiview images or a video of the reference head, thus restricting their usage. Moreover, rendering speeds for NeRF-based models typically do not reach the levels needed for real-time applications.

Real-time performance with high rendering quality has been achieved through representations based on 3D Gaussians [17, 18, 20, 23, 30], whose positions, orientations, and densities are optimized for the reference head. By accounting for the visibility of each Gaussian and rendering only those that can be seen, real-time performance is attainable [24]. Rigging 3D Gaussians to a parametric head model allows them to be dynamically controlled through parameter manipulation. Our work utilizes this representation but controls face and head motion using VASA-1 motion latents, for which residuals to these Gaussians are incorporated to precisely model the subtle expression details that these latents capture.

**3D Head Reconstruction.** 3D head avatars can be reconstructed from a reference head using multi-view correspondences [17, 18]. For greater practical convenience, much attention has focused on one-shot methods that require only a single head image. These methods either rely on a parametric head model as a strong prior [8, 20] or predict a volumetric [9] or tri-plane [14, 15, 19] representation for NeRF rendering. In this work, we propose to leverage the considerable recent advance in 2D talking face generation [1] to synthesize close approximations of additional views of the input face, providing further data for training.

A collection of frames synthesized in our method may resemble monocular video input, which is used in several works for 3D head avatar reconstruction [4, 5, 6, 7, 10, 11, 12, 13, 16]. However, our training data differs significantly from monocular video in two respects. One is that a broad range of head poses and facial expressions can be synthesized with our approach, much beyond than what can reasonably be captured in a video of a reference head. The other is that the images generated by VASA-1 [1] lack temporal texture consistency, which creates problems when using common training losses based on pixel-wise comparisons. We overcome this issue through judicious selection of losses that are robust to such artifacts.

**Head Avatar Animation.** Head avatars are typically modeled in a way that they can be driven using parameters of parametric models like 3DMM and FLAME [4, 5, 6, 7, 8, 9, 10, 11, 12, 13, 14, 15, 16, 17, 18, 19, 20]. Their reliance on parametric models for animation encoding and control limits the expressiveness of faces. In contrast, our method drives animation using VASA-1 motion latents, which provide a richer expression representation learned from an abundance of 2D head videos. Though the 3D Gaussian splats that represent head shape in our work are rigged to a FLAME model, we incorporate residuals conditioned on the motion latents, giving expression control to these latents.

## 3 Method

Our VASA-3D framework, illustrated in Fig. 2, is built on two main ideas: adapting the VASA-1 motion latent to 3D, and leveraging the high realism of VASA-1 [1] in 2D talking head video generation to facilitate single-shot customization of the 3D head model. We train VASA-3D models with carefully-designed losses that enhance visual quality while avoiding issues that may arise from the synthesized videos.

### 3.1 VASA-3D Model

**3D Gaussian Representation.** Our VASA-3D model is based on 3D Gaussians [24] equipped with Gaussian deformation fields driven by the VASA-1 motion latent. The 3D head is represented as a set of 3D Gaussians $\mathcal{G} = \{\mathbf{g}_i = (\boldsymbol{\mu}_i, \boldsymbol{r}_i, \boldsymbol{s}_i, \boldsymbol{c}_i, \alpha_i)\}_{i=1}^{N}$, each with position $\boldsymbol{\mu}$, rotation $\boldsymbol{r}$, scale $\boldsymbol{s}$, color $\boldsymbol{c}$, and opacity $\alpha$. To ease learning of Gaussian parameters and deformation fields, we make use of priors from existing 3D head parametric models. Specifically, we use 3D Gaussians bound to a

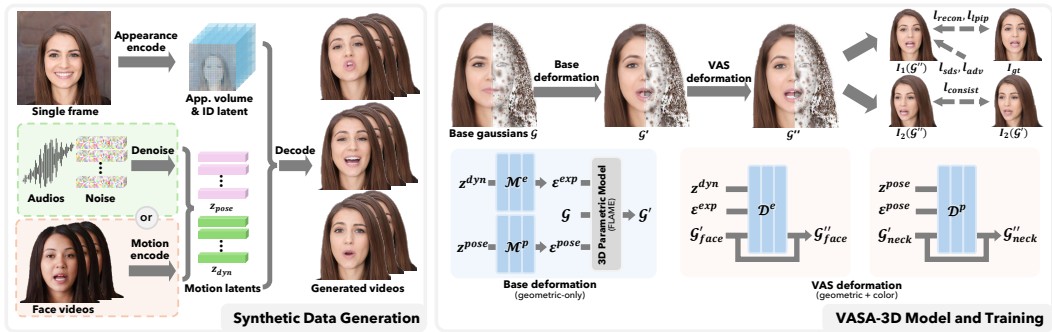

Figure 2: Overview of VASA-3D. Given a single portrait image, we use the VASA-1 model to generate a collection of synthetic talking face videos as well as their corresponding motion latents, which are used to train a VASA-3D model. The driving sources for these videos can be in-the-wild audios and/or face videos. Our VASA-3D model is represented by deformable 3D Gaussians attached to a FLAME mesh. Two deformation fields are applied to the Gaussians, one based on the FLAME mesh and another modulated by VASA motion latents. After training, a VASA-3D model can be driven with VASA motion latents generated from audios or videos in real time.

FLAME model [31], a representation introduced in GaussianAvatars [17]. Unlike in [17] and other previous works, animations will be driven by the VASA-1 latent.

We decompose the deformation into two parts: a *Base Deformation*, and another deformation we call *VAS Deformation*. The former is driven by the FLAME parametric model to change the geometric properties of the Gaussians including position, rotation, and scale. The latter derives the fine-grained geometric and color variations, which is crucial for expressing the motion nuances captured in VASA-1 and improving the rendering quality.

**Base Deformation.** Given a motion latent $\mathbf{x} = [\mathbf{z}^{dyn}, \mathbf{z}^{pose}]$ produced by the VASA-1 diffusion model, we first map them to FLAME parameters using two MLPs. The first MLP $\mathcal{M}^e$ converts the facial dynamics code $\mathbf{z}^{dyn}$ to the expression-related FLAME parameters $\varepsilon^{exp} = (\boldsymbol{\psi}, \boldsymbol{\theta}^{eye}, \boldsymbol{\theta}^{jaw})$ which includes the expression PCA coefficients, eye pose, and jaw pose, respectively. The second MLP $\mathcal{M}^p$ uses $\mathbf{z}^{pose}$ to predict the FLAME pose parameters $\varepsilon^p = (\boldsymbol{\theta}^{neck}, \boldsymbol{\theta}^{global}, \mathbf{t})$ including neck rotation, global rotation, and global translation. These two mappings can be written as:

$$\varepsilon^{exp} \leftarrow \mathcal{M}^e(\mathbf{z}^{dyn}), \tag{1}$$

$$\varepsilon^{pose} \leftarrow \mathcal{M}^p(\mathbf{z}^{pose}). \tag{2}$$

Both $\mathcal{M}^e$ and $\mathcal{M}^p$ have three fully connected layers, with 256 hidden units per layer followed by a ReLU activation function. Additionally, a shape coefficient $\varepsilon^{shape}$ is jointly optimized during training and fixed during inference.

Given these parameters, the FLAME mesh will be rigged accordingly, which drives the changes of $(\boldsymbol{\mu}_i, \mathbf{r}_i, \mathbf{s}_i)$ for the Gaussians $\mathbf{g}_i$ attached to the mesh triangles. Additional details of this process can be found in [17].

**VAS Deformation.** We further learn dense Gaussian deformation fields for our 3D head model, modulated by VASA-1 motion latents. Two MLPs are introduced to predict the deformations of Gaussians in the face and neck region, respectively. The first MLP $\mathcal{D}^e$ takes Gaussians $\mathbf{g}_i$ in FLAME's facial region $\Omega_{face}$ as well as the VASA-1 facial dynamics latent $\mathbf{z}^{dyn}$ as input and predicts the full transformation $\Delta \mathbf{g}_i = (\Delta \boldsymbol{\mu}_i, \Delta \mathbf{r}_i, \Delta s_i, \Delta \mathbf{c}_i, \Delta \alpha_i)$. We also feed the FLAME expression parameters $\varepsilon^{exp}$ to the MLP so that it is aware of the current base expression. The second MLP is responsible for the FLAME neck region $\Omega_{neck}$. It takes the Gaussians, VASA pose $\mathbf{z}^{pose}$ and FLAME pose parameters $\varepsilon^{pose}$ as input to predict the residuals. The VAS deformations can be expressed as:

$$\Delta \mathbf{g}_{i \in \Omega_{face}} \leftarrow \mathcal{D}^e(\mathbf{g}_i, \mathbf{z}^{dyn}, \varepsilon^{exp}), \tag{3}$$

$$\Delta \mathbf{g}_{j \in \Omega_{neck}} \leftarrow \mathcal{D}^p(\mathbf{g}_j, \mathbf{z}^{pose}, \varepsilon^{pose}). \tag{4}$$

These two MLPs share the same architecture as $\mathcal{M}^e$ and $\mathcal{M}^p$ except for the different input and output dimensions. For all input Gaussian positions $\boldsymbol{\mu}$, we apply sinusoidal positional encoding with $L = 4$ [29].

**Animation and Rendering.** Once trained, VASA-3D models can be animated using VASA-1 motion latents. The driving sources can be either audios or videos. For audio input, we use VASA-1's diffusion transformer to generate motion latents. For videos, we use VASA-1 motion encoders for latent code extraction. The animation frames are efficiently rendered with Gaussian Splatting and the whole animation and rendering pipeline can run in real time on a commodity GPU.

## 3.2 Synthetic Training Data Generation

We leverage VASA-1 to generate video frames with a diverse set of poses and expressions from the given single image. To achieve this, one can either use real speech audios and/or face videos for data generation. For example, in most of our experiments, we randomly sample up to 10 hours of video clips from the VoxCeleb2 dataset [32] to render the training data. We extract the VASA-1 motion latent for each frame and use the VASA-1 decoder to drive the portrait image and synthesize the corresponding frames. The paired motion latent and video frame data will be used for VASA-3D model training, which we present next.

## 3.3 Robustified Model Training

We train our models in an end-to-end manner where all the trainable modules, including Gaussian parameters and the MLPs for deformation, are trained together from scratch.

*Challenges.* Our synthesized training data and the dense free-form deformation in our 3DGS-based head model give rise to several challenges for training.

- Unlike real videos, inconsistency of temporal texture and facial shape exists among the synthesized frames.
- Large viewing angles are often missing from the training data, leading to difficulties in shape reconstruction.
- The inclusion of residuals for the Gaussians can lead to overfitting to the training video frames.

We apply the following losses to train the model effectively without succumbing to these issues.

**Reconstruction Losses.** We use a combination of the structural similarity index measure (SSIM) and the $L_1$ color difference as the photometric loss between the generated image and the ground truth image:

$$L_{recon} = \lambda_{ssim}L_{ssim} + (1 - \lambda_{ssim})L_1. \tag{5}$$

**Perceptual Losses.** As temporal texture inconsistency can reduce the efficacy of the photometric loss, we rely on perception-level losses that measure visual quality but are robust to temporal inconsistency among the training frames. Specifically, we apply the Learned Perceptual Image Patch Similarity (LPIPS) loss [33] with a pretrained VGG network [34]. To further improve realism, we add multi-scale patch discriminators and apply an adversarial loss for training. We employ three discriminators with different input image scales and trained them together with our model. The perceptual loss functions can be written as:

$$L_{perc} = \lambda_{lpips}L_{lpips} + \lambda_{adv}L_{adv}. \tag{6}$$

**SDS Loss.** Since the synthesized video data generally covers a limited range of poses, we apply the SDS loss [35] to minimize visual artifacts in side views and to enlarge the range of valid viewing angles. Specifically, we render our model from random viewing angles for which we apply the SDS loss $L_{sds}$. Random views are uniformly sampled from azimuth angles in the range $[-180°, 180°]$ and elevation angles in the range $[-22.5°, 22.5°]$. We use StableDiffusion v2.1 [36] as the diffusion model in our SDS loss with classifier-free guidance factor 10.0 and gradient scale 0.001. The text prompt is 'human portrait, realistic photography, by DSLR camera'.

Though the dense VAS deformations help in capturing detailed expressions and improving image quality, they also require careful design of regularization to avoid overfitting to each frame's training data. In light of this, we compute $L_{recon}, L_{per}$ and $L_{sds}$ for the rendered images of the Gaussians *both after base deformation and after VAS deformation*, denoted as $\mathcal{G}'$ and $\mathcal{G}''$, respectively. This

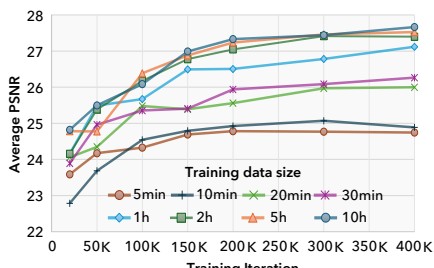

Figure 3: Effects of dataset size and training iterations

| Setting | PSNR↑ | L1↓ | SSIM↑ | LPIPS↓ | $S_C$↑ | $S_D$↓ |
|---|---|---|---|---|---|---|
| Basic | 25.74 | 0.0228 | 0.8544 | 0.0768 | 6.6347 | 8.1265 |
| +VAS deform. | 27.19 | 0.0195 | 0.8654 | 0.0695 | **6.9636** | **7.9050** |
| +$L_{sds}$ | 27.23 | 0.0195 | 0.8653 | 0.0707 | 6.9581 | 7.9191 |
| +$L_{consist}$ | **27.33** | **0.0192** | **0.8672** | 0.0706 | 6.9429 | 7.9221 |
| +$L_{cas}$ | 26.62 | 0.0209 | 0.8472 | **0.0657** | 6.9145 | 7.9422 |
| GT | – | – | – | – | 7.1517 | 7.7770 |

Table 1: Quantitative results of ablation studies

approach aims for $\mathcal{G}'$ to effectively capture facial features shared across frames to the extent possible through multi-frame consistency, while $\mathcal{G}''$ focuses on modeling the residual facial details of different frames.

**Render Consistency Loss.** Although the SDS loss helps to eliminate artifacts in profile regions, we found that it also tends to smooth out the details across all regions in our case. This issue is pronounced for $\mathcal{G}''$ because the Gaussian residuals are learned for each frame. Such flexibility makes $\mathcal{G}''$ susceptible to the side effect of the SDS loss, especially for regions not well captured by the current view (hence less constrained by the ground-truth reference image). In contrast, $\mathcal{G}'$ are less prone to this issue, as these Gaussians are optimized to jointly fit the multi-frame data with different poses and thus are less affected by $L_{sds}$. Therefore, we design a loss to regularize $\mathcal{G}''$ with $\mathcal{G}'$. Specifically, for each training iteration we render an additional pair of images with $\mathcal{G}'$ and $\mathcal{G}''$ respectively, under a new view angle significantly different from the current training view. We apply a render consistency loss $L_{consist}$ between these two images $I'(\mathcal{G}'')$ and $I'(\mathcal{G}'')$ as

$$L_{consist} = LPIPS\big(I'(\mathcal{G}''), stop\_grad(I'(\mathcal{G}'))\big), \tag{7}$$

where the stop-gradient operator prevents $G'$ from being negatively affected.

In practice, the new view is randomly sampled with the azimuth angle in the range $[-35°, -55°]$ or $[35°, 55°]$ and elevation angle in the range $[-15°, 15°]$. Of the two azimuth ranges, we select the one that is farther away from the view of the training frame.

**Sharpening Loss.** To further increase the sharpness of the rendered results, we *optionally* apply a contrast-adaptive-sharpening (CAS) filter [37] to the model's rendered images and use them to further train the model. Specifically, we apply the CAS filter to the rendered image and then apply the LPIPS loss between the sharpened image and the original image. The CAS loss $L_{cas}$ is applied at the end of the training process as a lightweight finetuning step.

**The overall loss function** is a combination of the aforementioned losses:

$$L = L_{ssim} + L_1 + L_{lpips} + L_{adv} +$$
$$L_{sds} + L_{consist} + L_{cas} + L_{others}, \tag{8}$$

where the loss weights are omitted for brevity, and $L_{others}$ stands for other loss functions introduced in [17] such as the Gaussian position and scale losses (see [17]). More details about our losses and their implementation can be found in the *supplementary material*.

## 4 Experiments

### 4.1 Analysis and Ablation Study

We conduct experiments to analyze our method as well as the design choices. In the following experiments, we use ten portraits, including five males and five females, generated by StyleGAN2 [2] to train our models. We train the models using 4 NVIDIA A100 40G GPUs and a batch size of 4. A $512 \times 512$ resolution is used for both the training data and VASA-3D rendering throughout this paper.

**Inference Speed.** Given an audio clip as input, the animation and $512 \times 512$ video frame rendering of our VASA-3D model can *run at **75fps** with a preceding latency of only 65ms*, evaluated on a single NVIDIA RTX 4090 GPU. A real-time demo is provided in the supplementary videos.

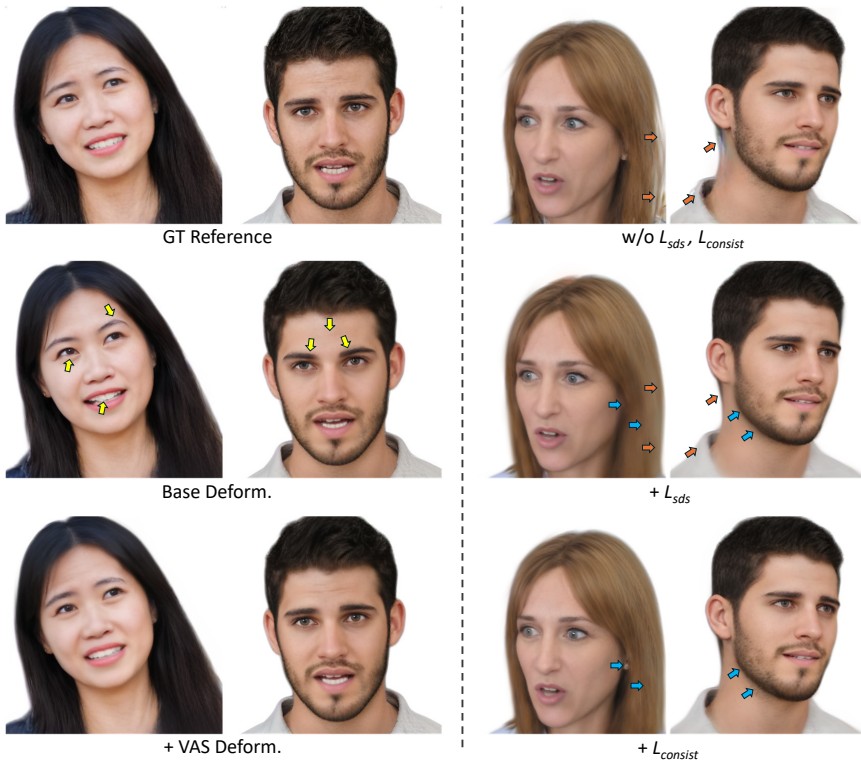

Figure 4: Left: VAS deformation not only improves image quality (see also Table 1) but also captures facial nuances subtle yet critical for expressing emotions. Right: The SDS loss eliminates artifacts in profile regions while the render consistency loss enhances the details smoothed out by the SDS loss.

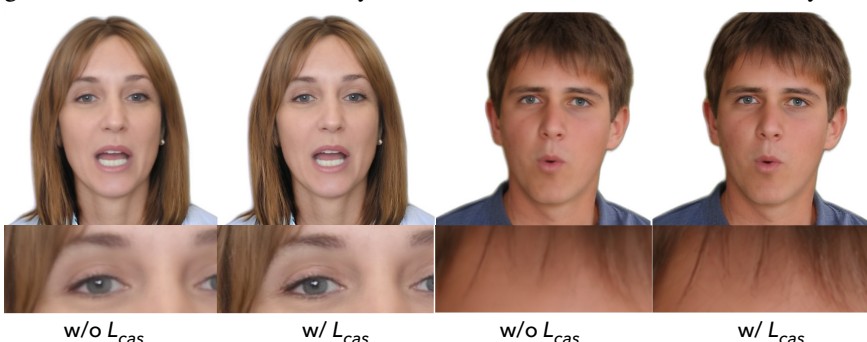

Figure 5: The CAS loss improves the overall rendering sharpness.

### 4.1.1 Dataset Size and Training Time

We first analyze the influence of dataset size (*i.e.*, length of training videos synthesized by VASA-1) and training time (in iterations) on result quality. For each image, we use VASA-1 to render eight training datasets of different sizes, *i.e.*, 5min, 10min, 20min, 30min, 1h, 2h, 5h, 10h, using VASA-1 latents extracted from random video clips in VoxCeleb2 [32]. We evaluate the models at varying training iteration numbers (up to 400K) on our test set, which are VASA-1 generated videos of 3min for each image.

Figure 3 shows how the average PSNR scores across the ten portraits on the test set improves as the dataset size and training iterations increase. It is observed that the improvements almost plateau after the dataset size reaches 2 hours and after the number of iterations exceeds 200K. Therefore, we set the total number of iterations to 200K by default in the following experiments. We also trained our model with 20K iterations and compare it against other baselines in some experiments below. Since the training data size does not affect training time, we simply set it to 10 hours. For each portrait, a 10-hour dataset can be rendered in less than 1 hour on 4 NVIDIA A100 40G GPUs. Training with 20K/200K iterations takes about 1.8/18 hours for each model.

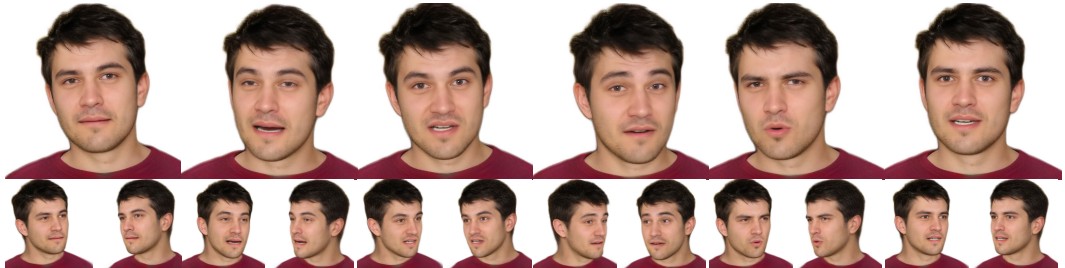

Figure 6: Example frames from the generation results of VASA-3D. The first row shows the frontal view and the second row presents side views of the same frames.

|  | FID↓ | $S_C$↑ | $S_D$↓ | ID Sim↑ |
|---|---|---|---|---|
| VASA-1 | 5.24 | 8.142 | 6.9237 | 0.8154 |
| VASA-3D | 7.45 | 8.121 | 6.9300 | 0.7874 |

Table 2: Audio-driven generation comparison with VASA-1, the results of which represent our *upper-bound performance* as our training data is generated by it. VASA-3D has only a subtle performance gap to VASA-1, yet providing true-3D, freeview-renderable talking heads not achieved by VASA-1.

Some examples of our results generated by the default setting are presented in Fig 6. Our method is shown to generate high-quality 3D head renderings with accurate audio-lip sync, vivid facial expressions, and lively head motions. Video results are provided in the *supplementary materials*, where readers can more fully examine the quality of VASA-3D generation.

### 4.1.2 Effect of VAS Deformation

We further train different variants of our method with the same training and test setup, and the quantitative results are presented in Table 1, where the evaluation metrics include PSNR, L1 error, SSIM, LPIPS, and lip-audio synchronization score. For the lip-sync score, we use SyncNet [38] to assess the alignment confidence score $S_C$ and feature distance $S_D$.

Table 1 shows the effect of our VAS Deformation, compared to a basic setting with Base Deformation only. All the image quality and lip-sync metrics are significantly improved, demonstrating the importance of the VASA-latent-driven Gaussian deformation. Some visual comparisons are presented in Fig. 4. The image quality is clearly improved by VAS deformation. More importantly, the facial expressions including subtle facial details follow the ground-truth reference frame more closely, displaying the expressive talking features with nuanced facial details that are modeled by VASA-1.

### 4.1.3 Effects of Different Losses

Fig. 4 exhibits improvements with the SDS loss and render consistency loss. Due to the limited pose coverage of our training data, artifacts can be clearly observed for the results without SDS loss under side views rendered at $45°$ azimuth angles. The SDS loss provides additional regularization to the model and eliminates the artifacts in side views. However, it also tends to smooth out details. Adding the render consistency loss improves the results by enhancing the rendering details. Fig. 5 shows results from training with the CAS loss, where the images are further sharpened.

Table 1 shows numerical results with different losses. The full method without the CAS loss (*i.e.*, the fourth row) yields the best image quality in terms of the PSNR, L1 and SSIM metrics, whereas the method with the CAS loss has the best perceptual score measured by LPIPS due to the enhanced image sharpness. The lip-sync scores remain stable with different loss functions incorporated.

## 4.2 Audio-Driven Generation and Comparisons

In this experiment, we construct our training datasets by collecting five random audio clips from the web (two males and three females), each with a total length of 25 minutes[2]. For each audio clip, we

---

[2]Our model can use either video frames or audios for training. Here we use audios since some compared methods require continuous videos and cannot utilize the 10-hour dataset in Sec. 4.1 containing short video clips.

| | $S_C\uparrow$ | $S_D\downarrow$ | ID Sim$\uparrow$ | *US* - Video Quality$\uparrow$ | *US* - Overall Preference$\uparrow$ |
|---|---|---|---|---|---|
| ER-NeRF | 5.921 | 8.7788 | 0.7732 | 1.82 | 1.08% |
| GeneFace | 5.922 | 9.6066 | 0.7857 | 1.73 | 0.72% |
| MimicTalk | 5.270 | 10.9368 | 0.7748 | 2.23 | 3.58% |
| TalkingGaussian | 6.701 | 8.1061 | **0.7971** | 2.38 | 0.72% |
| VASA-3D | **8.121** | **6.9300** | 0.7874 | 4.29 | 93.91% |
| VASA-3D (20k iter) | 7.980 | 7.0020 | - | - | - |

Table 3: Comparison with audio-driven 3D talking head methods that are trained on videos. Note that all methods except ours do not generate head pose. "*US*" denotes User Study (see text for details). VASA-3D (20k iter) denotes our model trained with only 1/10 of default iteration number.

apply VASA-1 to drive a StyleGAN2 portrait image to generate the synthetic video data. We only use the first *20-minutes* to train the models, and the remaining 5-minutes are used as the test set.

**Comparison with VASA-1.** We compare our method to VASA-1, our training data generator, to check the quality difference of the generated talking videos. Table 2 presents the frame FID [39], LipSync [38] confidence score $S_C$ and feature distance $S_D$, and the facial identity similarity between test video frames and driving portrait images, calculated with ArcFace [40]. The performance gap between VASA-3D and VASA-1 is small.

**Comparison with Previous 3D Talking Head Avatar Methods.** To our knowledge, no existing method deals with the same task as ours – *i.e.*, single photo to expressive, fully-animatable 3D head avatar driven by audio – making the comparison difficult. Most audio-driven 3D talking avatars are trained on long videos, and they typically do not generate full head dynamics such as head pose.

Still, to facilitate comparison with state-of-the-art techniques *for reference purposes*, we consider the following methods: ER-NERF [41], GeneFace [42], MimicTalk [16], and TalkingGaussian [23]. We employ the same video data as in our VASA-3D to train these methods and use the audios from the test set to generate videos. Table 3 shows the evaluation results of different methods. Our model surpasses the other methods on the LipSync metrics by a wide margin. Its identity similarity score is marginally lower than TalkingGaussian [23] and better than others. We further conduct user studies to evaluate the rendered video quality and the overall realism of the audio-driven results. 15 participants were invited to assess: 1) the visual quality of the rendered talking head videos (audio muted), with ratings from 1 to 5, and 2) the user preference of the results from the five compared methods, judged by overall realism. Our visual quality rating was significantly higher than the other methods and the users chose the VASA-3D as the best one for 93.91% of the presented cases.

**More comparisons.** We further compare our method with related works on video-driven 3D talking head animation (*i.e.*, the face reenactment task). Note that face reenactment is not a focus our work; the goal here is to further compare VASA-3D's rendered video quality as well as its expressiveness on facial expression against prior art. Details of this experiment can be found in the *suppl. material*.

### 4.3 Generation with Additional Control Signal

Inheriting the capabilities of VASA-1, our VASA-3D can take additional control signals besides an audio clip, such as eye gaze direction, head distance, and emotion offset. Fig. 7 as well as our supplementary video present typical results with emotion offset control, where the generated 3D talking heads closely adhere to different emotion offsets and exhibit emotive talking styles.

### 4.4 Artistic Image Experiments

We also tested our method on artistic-style portrait images, with some examples shown in Fig. 1 and Fig. 8. Our method can effectively handle such artistic images and produce convincing 3D videos.

## 5 Conclusion

VASA-3D offers unparalleled realism for audio-driven 3D head avatars by presenting a way to leverage the extensive expression data present in online 2D head videos. With a meticulously designed architecture and training scheme, our model can be easily customized using only a single

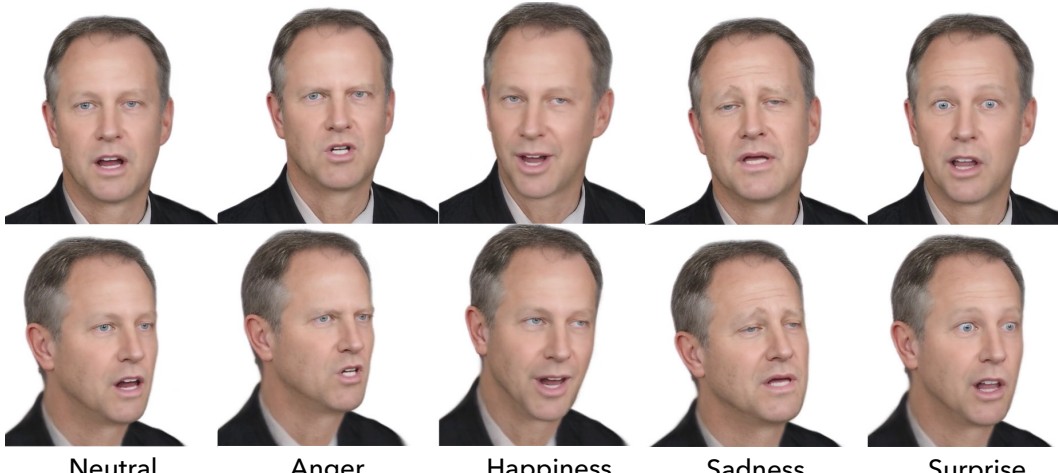

| Neutral | Anger | Happiness | Sadness | Surprise |

Figure 7: Audio-driven generation results with additional control signal of emotion offset. The results are generated with the same audio clip. *See the accompanying video for animated results with audio.*

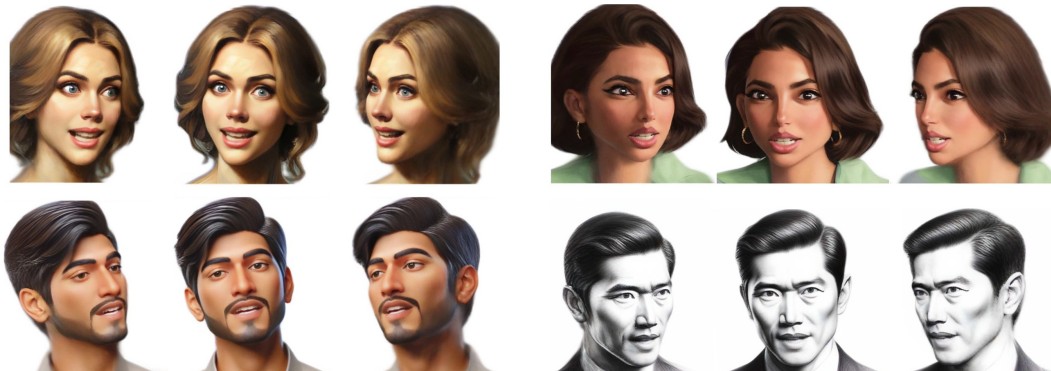

Figure 8: Results on artistic-style images. *See our videos for animated results with audio.*

portrait image. We believe our approach paves the way for more immersive and engaging virtual experiences with 3D head avatars.

**Limitations and Future Work.** Our method still has several limitations. Limited by the viewing angles of the synthetic training videos, it does not model the back of heads. This issue could potentially be resolved through 3D inpainting, since the back of a head is mostly rigid. Similar to VASA-1, our method does not handle dynamic elements such as accessories. Extending VASA-3D to include the upper body is another interesting direction we will explore in future.

## 6   Societal Impacts and Responsible AI Considerations

Our research aims to support positive applications of virtual AI avatars and is not intended for creating misleading or deceptive content. However, like other related techniques, VASA-3D could potentially be misused in generating the likeness of a real person. Throughout the development of VASA-3D, responsible AI considerations were factored into all stages. To safeguard against such harm, we are training face forgery detection models that incorporate our models' outputs as part of the training data. Though VASA-3D produces visually realistic results, we have found that they are easily distinguishable from authentic videos by these models and improve the models' generalizability.

While recognizing the potential for misuse, it is important to acknowledge the substantial positive impact that our research technique could eventually have. We are currently examining potential benefits, such as its application in an AI coworker and AI tutor, which can enhance latent intelligence accessibility for knowledge workers and learners. These applications highlight the significance of this research and other related investigations. We are committed to developing AI responsibly, with the goal of advancing human well-being.

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

# A  More Training Details

For the SDS loss $L_{sds}$, we apply it every 10 iterations due to its high computation cost. We apply $L_{sds}$ with time step $t$ in a normalized range of $t_{min} = 0.02$ to $t_{max} = 0.98$, with $t_{max}$ decaying by 0.98 every 2,000 iterations. The CAS loss $L_{cas}$ is applied after 200K iterations, and the model is fine-tuned for an additional 20K iterations with $L_{cas}$ and other losses.

In all our experiments, the loss weights are set as $\lambda_{ssim} = 0.1$, $\lambda_{lpips} = 1.0$, $\lambda_{adv} = 0.001$, $\lambda_{sds} = 1.0$, $\lambda_{consist} = 0.01$, and $\lambda_{cas} = 10.0$. These weights are empirically chosen without careful tuning.

As mentioned in Sec. 4.1 of the main paper, our models are trained for 200K iterations by default, excluding the CAS loss finetuning iterations. Gaussian densification and pruning start at the 10K iterations, with intervals of 2K iterations. We stop this process after 100K iterations or when the total number of Gaussians exceeds 200,000.

# B  More Experimental Results

**Comparison with 3D Talking Head Avatar Methods.**  Sec 4.2 of the main paper compared our method against some 3D talking head avatar models, with numerical results provided including user studies. Fig.A presents some examples. Our method is shown to generate high-quality 3D head renderings with accurate audio-lip sync, vivid facial expressions, and lively head motions, surpassing the capabilities of existing 3D talking head avatar methods.

Fig. B shows the screenshot of our user study interface. To assess the visual quality of the rendered videos, we asked the participants to assign satisfaction scores from 1 to 5. Videos were presented one at a time, with the play order of different methods randomized for each test case. We asked the participants to provide their own judgment of satisfaction when watching a talking avatar on screen. Note that individual satisfaction levels may vary; however, the averaged scores provide a fair basis for comparison as each participant rated results from all methods. To evaluate user preferences for overall realism, we display the results of all compared methods side by side and ask the participants to select the one that looks the most realistic to them. Method names remained anonymous and their orders are randomly shuffled for each test case.

**Comparison with 3D Face Reenactment Methods.**  As mentioned in Sec 4.2, we further compare our method with related works on video-driven 3D talking head animation, *i.e.*, the face reenactment task. Note that face reenactment is not the focus of our work; the goal here is to further check our video quality and its expressiveness on facial expression.

Specifically, we collect 26 portraits, each with 1-minute high-quality talking videos, from the CelebVHQ [43] dataset. For each portrait, we randomly selected one frame from the video and use the VASA-1 decoder to render 10 hours of training frames from it, with VASA-1 latents extracted from random VoxCeleb2 video clips. With the collected 1-minute real talking videos as test sets, we compared our model with the following video-driven 3D head avatar methods: GAGAvatar [20], GPAvatar [28], Real3DPortrait [19], Voodoo3D [44] and Portrait4D-v2 [45].

Table A shows the averaged results of these methods with different metrics. Our model outperforms all the other methods under all the metrics.

| | PSNR↑ | PSNR$_{Face}$↑ | L1↓ | SSIM↑ | LPIPS↓ | $S_C$↑ | $S_D$↓ |
|---|---|---|---|---|---|---|---|
| GAGAvatar | 25.74 | 30.53 | 0.0257 | 0.8695 | 0.0829 | 5.502 | 8.693 |
| GPAvatar | 24.91 | 29.41 | 0.0288 | 0.8583 | 0.1016 | 4.785 | 9.256 |
| Real3DPortrait | 23.78 | 28.04 | 0.0338 | 0.8481 | 0.1091 | 4.971 | 9.179 |
| Voodoo3D | 23.43 | 28.39 | 0.0343 | 0.8380 | 0.1209 | 4.307 | 9.500 |
| Portrait4D-v2 | 23.19 | 27.55 | 0.0356 | 0.8325 | 0.0946 | 5.823 | 8.530 |
| VASA-3D | **26.21** | **31.11** | **0.0255** | **0.8741** | **0.0760** | **6.453** | **7.996** |
| GT | – | – | – | – | – | 6.673 | 7.802 |
| VASA-1 | 25.93 | 31.01 | 0.0261 | 0.8544 | 0.0809 | 6.302 | 8.061 |

Table A: Comparison with video-driven 3D face reenactment methods.

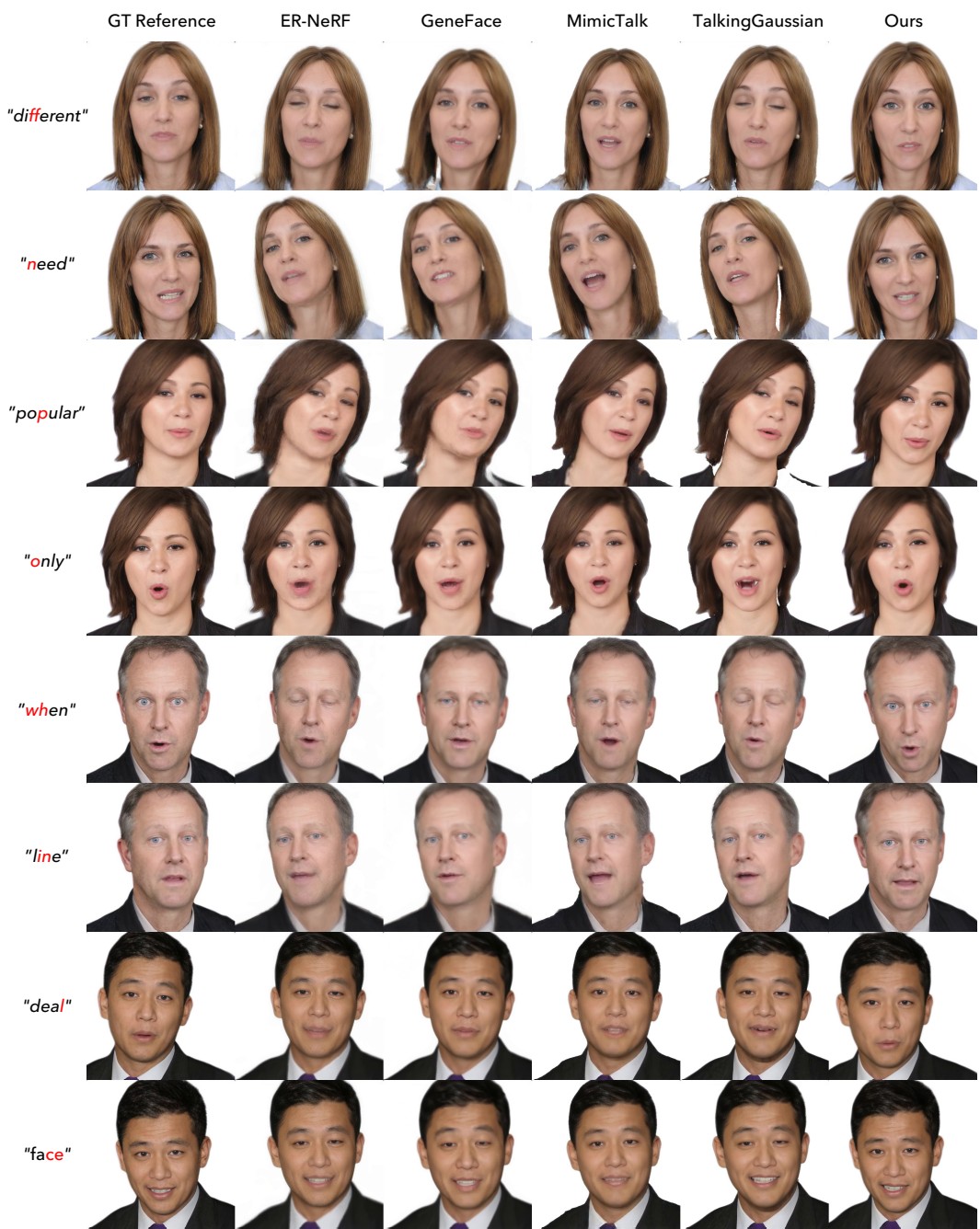

Figure A: Visual examples of audio-driven 3D talking head generation. **Note**: all methods except ours do not produce head pose, so we apply the pose sequences in training data for them. *Best viewed with zoom; see our supplementary video for comprehensive comparisons.*

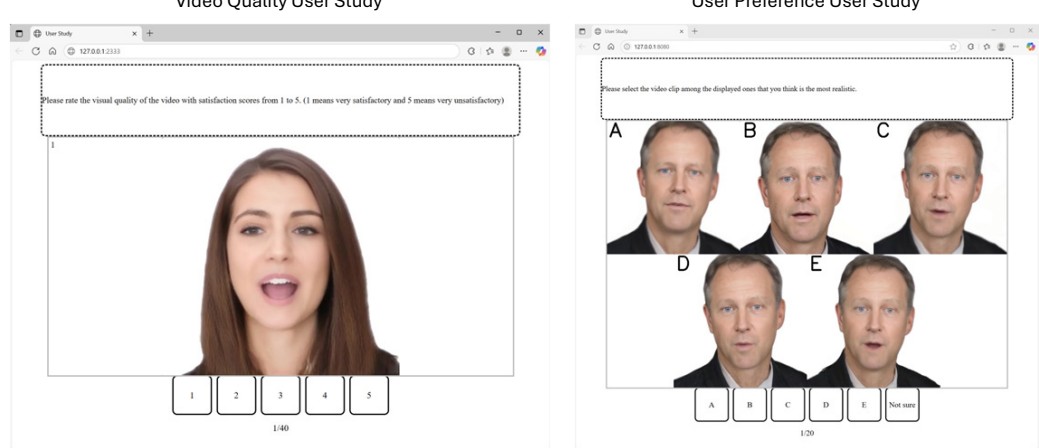

Video Quality User Study          User Preference User Study

Figure B: The user study interface for result comparison with existing audio-driven 3D talking head avatar methods. **Left:** To assess the visual quality of the rendered videos, we asked the participants to assign satisfaction scores from 1 to 5. Videos were presented one at a time, with the play order of different methods randomized for each test case. We asked the participants to provide their own judgment of satisfaction when watching a talking avatar on screen. Note that individual satisfaction levels may vary; however, the averaged scores provide a fair basis for comparison as each participant rated results from all methods.. **Right:** To evaluate user preferences for overall realism, we display the results of all compared methods side by side and ask the participants to select the one that looks the most realistic to them. Method names remained anonymous and their orders are randomly shuffled for each test case.

