# OpenReview forum: "VASA-3D: Lifelike Audio-Driven Gaussian Head Avatars from a Single Image"
_NeurIPS.cc/2025/Conference — NeurIPS 2025 poster_

### Official Review · Reviewer_mvUV · 2025-07-01

**Clarity:** 3
**Significance:** 2
**Originality:** 2
**Rating:** 4
**Confidence:** 3

**Summary:**

This paper proposes VASA-3D, an audio-driven framework for generating expressive 3D head avatars from a single image. It builds on the motion latent of VASA-1 and maps it to a 3D head model through an optimization process using self-synthesized multi-view data. The method achieves realistic 3D talking heads and supports real-time free-viewpoint rendering at 512×512 resolution.

**Questions:**

1. As VASA-1 has not been open-sourced to date, and this work is essentially a 3D extension built upon it, it raises the question of whether the authors plan to release the code and models for VASA-3D. Open-sourcing this work would be particularly important for reproducibility and for promoting further research in the field.
2. The paper adopts the motion latent from VASA-1 to drive the 3DGS representation. However, it is unclear why this latent space is preferred over directly using FLAME parameters for motion control, especially given that the base deformation in the proposed method is essentially obtained by mapping the VASA-1 latent to FLAME parameters via MLPs. The authors should clarify what advantages the VASA-1 latent offers compared to FLAME parameters.
3. In the VAS deformation module, the model learns residuals for all 3DGS parameters. As discussed in the paper, this may lead to overfitting to the synthesized training data. Would it be possible to improve generalization by only learning animation-relevant parameters such as position and rotation, while keeping other attributes like color, scale, and opacity fixed?
4. The training data for this method is generated using VASA-1 to produce over 10 hours of video. It is unclear whether pose control is applied during this process to ensure diverse viewpoints. Without explicit pose variation, audio-driven generation may result in overly redundant, front-facing video frames. The authors are encouraged to provide a more detailed description of the video generation pipeline.
5. Does the FLAME model used in this work include teeth geometry? Additionally, the paper should clarify how the 3D Gaussian Splatting (3DGS) representation is bound or aligned with the FLAME mesh. It would also be helpful to specify the total number of Gaussians used for rendering the final avatar.

**Ethical Concerns:**

["NO or VERY MINOR ethics concerns only"]

**Final Justification:**

The authors have addressed my concerns, and I will raise my rating accordingly.

**Limitations:**

yes

**Quality:**

3

**Strengths And Weaknesses:**

### Strengths
1. VASA-3D effectively bridges 2D and 3D head modeling by leveraging the motion latent of high-quality 2D talking heads (VASA-1) and translating it into 3D space.
2. The paper presents experiments demonstrating that VASA-3D outperforms recent methods in both visual quality and motion realism.

### Weaknesses
1. VASA-3D is a speech-driven 3D avatar generation method built upon 3D Gaussian Splatting (3DGS), which plays a central role in the framework, including learning 3DGS parameters from a reference image and audio. However, the paper lacks a comprehensive discussion of prior work on 3DGS and its applications in human avatar modeling. Such as GaussianAvatars and GAGAvatar (mentioned in the paper) are not sufficiently reviewed in the related work section.
2. Although the paper focuses on audio-driven 3D head avatar generation and claims that few existing methods address this specific setting, it is worth noting that VASA-1 can also be driven by video. There are several recent methods capable of generating video-driven 3D avatars, such as FlashAvatar[1] and GBS[2]. However, the paper does not include experimental comparisons with these relevant approaches. Including such comparisons would provide a more complete evaluation and help clarify the strengths and limitations of the proposed method in the broader context of 3D avatar generation.
3. This work is largely built upon the foundation of VASA-1. Its main contribution lies in constructing a prediction model that maps VASA-1’s motion latent to 3DGS parameters, and optimizing it through a set of loss functions. However, both the MLP-based prediction model and the loss formulations are commonly used in prior work, limiting the novelty of the proposed method.
4. There are some typos in the paper. For example, in line 212, the symbol following “two images” is inconsistent with the one used in Equation (7).

[1]. Jun Xiang, Xuan Gao, Yudong Guo, and Juyong Zhang. Flashavatar: High-fidelity head avatar with efficient Gaussian embedding. In The IEEE Conference on Computer Vision and Pattern Recognition (CVPR), 2024
[2].Shengjie Ma, Yanlin Weng, Tianjia Shao, and Kun Zhou. 3d Gaussian blendshapes for head avatar animation. In ACM SIGGRAPH 2024 Conference Papers, pages 1–10, 2024

---

> ### Author Rebuttal · Authors · 2025-07-29
>
> Thank you for the valuable comments and suggestions. Before we address the reviewer's questions, we'd like to summarize the motivation and main contribution of our paper below.
>
> **Motivation, Contribution, and Relationship to Video-Driven Methods:**
>
> In our work, we observed that existing audio-driven 3D talking head methods cannot achieve highly realistic and vivid talking head results. The quality is far worse than the 2D talking face counterpart. After examination, we found that most existing methods apply 3D parametric models, in particular, FLAME, for expression control, following a generic "audio->FLAME expression->deformed 3D model" paradigm. However, existing 3D parametric models like FLAME are constructed by registering 3D scans – typically from just a few hundred subjects and without sophisticated expression design – and then applying PCA. They have intrinsic limitations in handling nuanced motion and subtle expressions of real human faces when talking and expressing emotions. So we propose to use a latent space learned on massive 2D face videos to drive 3D avatars, an attempt not made by any previous methods. We also propose a framework to map the learned latents to 3D Gaussian variations (Sec. 3.2) and a set of carefully designed losses to handle the training on synthesized data and addressing the challenges they incur (Line 170).
>
> We have demonstrated the clear advantage of our method compared to using FLAME parameters. We also achieved high-quality and naturalistic audio-driven 3D talking head animation results from a single image, significantly pushing the state-of-the-art quality in this domain and minimizing the performance gap to audio-driven 2D talking head methods.
>
> We also would like to re-emphasize that the task we handle in this work is *audio-driven talking face* generation. We acknowledge that there is rapid progress in 3D avatar reconstruction and video-based reenactment, such as those works pointed out by the reviewer. However, please note that audio-driven and video-driven 3D head avatar generation are often treated as distinct problems. Adapting video-driven methods to the audio-driven scenario is not trivial and achieving high realism and vividness in the talking scenario introduces challenges not fully addressed by these methods. Taking our base model GaussianAvatars [17] as an example, their video results on expression transfer (with FLAME expression codes) look convincing (see demos on their website). However, when it comes to *audio-synchronized motion behaviors and the expression of facial nuances subtle yet critical for expressing emotions*, FLAME's expression space is insufficient, as shown in our paper. In fact, these 3D avatar reconstruction and video-based animation methods represent technical advancements orthogonal to ours and they can be applied as our base model, similar to GaussianAvatars.
>
> ---
>
> We now respond to the reviewer's comments below
>
> **W1:** Due to the sheer number of papers on this topic, we opted to discuss the previous work based on the properties they share, rather than discuss each of them individually. For example, the 3D Gaussian representations used by GaussianAvatars and GAGAvatar are mentioned in Line 81-83 of the Related Work. For 3D head reconstruction, the multi-view correspondence used by GaussianAvatars is mentioned in Line 89-90, and the one-shot approach by GAGAvatar using a parametric head model as a strong prior is described in Line 90-92. For head animation, it is explained in Line 104-108 that GaussianAvatars and GAGAvatar utilize parametric models for animation encoding and control, while the animations of VASA-3D are instead driven by the VASA-1 motion latent.
>
> We feel that presenting the related work in this holistic manner allows us to position our method more clearly within the global context of the recent literature. However, to further address the reviewer’s concern, we could add the following at Line 106: *“For example, models based on 3D Gaussian splats such as GaussianAvatar [17] and GAGAvatar [20] control facial motion by modulating the expression parameters of the FLAME and 3DMM models, respectively, on which the splats are bound.”* Please let us know if this would be an acceptable solution.
>
> **W2:**  Our paper indeed focuses on *audio-driven* 3D head avatar generation as the reviewer noted. As we mentioned earlier, audio-driven and video-driven 3D head avatar generation are often treated as distinct problems. FlashAvatar and GBS are both video-driven 3D head avatar models, and they both use FLAME's expression parameters to drive the avatars, unlike ours that proposes using a latent space learned from massive 2D videos. These works are orthogonal to ours and can be potentially applied as our base model as well, similar to GaussianAvatars.
>
> **W3:** We wish to emphasize that the novelty of this work lies mostly in its new approach for driving animation of a 3D head avatar, rather than in the technical details for implementing this idea. Nevertheless, there also exist important technical innovations, as discussed below.
>
> Regarding our model framework, our instantiation is based on GaussianAvatars and we introduced a new module with two region-wise MLPs (neck and face) conditioned on VASA latent and head pose to regress additional Gaussian variations.
>
> Regarding our loss functions, they are designed to handle the unique challenges we had when training on the synthesized videos: temporal inconsistency, view angle limitation, and model overfitting (see Line 170-177). To our knowledge, no prior method has dealt with the same problem as ours. Although some loss terms might have been seen in different papers, our devised combination is a well-crafted and effective way to overcome the special challenges presented by our training data. Our novel-view render-consistency loss $L_{consist}$ with a $stop$_$gradient$ operator is a new one specially designed to regularize our final Gaussian output and mitigate the side effect brought by the SDS loss.
>
> **W4:** Thanks for pointing out the typos. We will correct them in the revised paper.
>
> **Q1:** Thank you for the suggestion regarding open source. We plan to release the code and some synthetic training data of VASA-3D for research purposes, which will facilitate checking the details of our VASA-3D implementations and exactly reproducing some typical VASA-3D avatars and the driving sequences shown in our paper.
>
> **Q2:** As we discussed above, our key motivation is that FLAME parameters are insufficient for capturing nuanced motion and subtle expressions of real human faces. This limitation is especially pronounced for the talking scenario where humans are sensitive to emotional expression and demand a high degree of motion realism. The latent space of VASA-1 is learned on a large volume of face videos to capture rich facial dynamics in a disentangled manner. The 2D talking face generation results of the VASA-1 also shows the capability of VASA-1 latent to represent nuanced motion and diverse talking behaviors. In light of this, we propose to apply it for 3D Gaussian head motion control and demonstrated its advantage in this work (Table 1, Fig. 3, etc.).
>
> Directly regressing dense 3D motions from VASA-1 latent vectors would pose significant challenges in learning. Therefore, we decouple the deformation to a base deformation guided by FLAME and a dense, free-from deformation (VAS deformation) guided by VASA-1 latent. This way, we properly leveraged the priors in FLAME to ease the learning but go beyond it to use VASA-1 latent as the ultimate driving signal.
>
> **Q3:** We tried this alternative approach on one case where we only predict position and rotation offsets in VAS deformation while keeping color, scale, and opacity fixed. A quality drop is observed as shown in the table below. And we did not observe obvious improvements related to generalization.
>
> | |PSNR$\uparrow$ | L1$\downarrow$ |SSIM$\uparrow$ | LPIPS$\downarrow$| $S_C\uparrow$| $S_D\downarrow$|
> | ---- |---- |---- |---- |---- |---- |---- |
> |  Pos-rot-only |  25.99 |0.0207 |0.8846 | 0.0665 | 6.918|7.926|
> |  Full |  26.54 | 0.0195 |0.8874 | 0.0645 |7.033  |7.884|
>
> **Q4:** Our training data can be generated using either audios or face videos. The 10-hour training data used to conduct ablation study and validate algorithm designs was generated using face videos  (Line 234). Specifically, we used video clips randomly sampled from VoxCeleb2 where each clip is typically of 5-10 seconds. The pose variation in the generated data is large. We also used audios to generate training data in Sec. 4.2 and 4.3 (Line 270, 292), where we use 20-minute continuous audios to generate training videos (some of the compared methods require continuous video for training and cannot utilize the collection of video clips in our 10-hour data). For this case, we did not apply additional pose augmentation for simplicity and the results look satisfactory. We conjecture that VASA-1-generated pose contains adequate variations and our loss designs are effective to facilitate the reconstruction with limited angles. Applying pose variations is nevertheless an excellent idea, and we thank the reviewer for this suggestion!
>
> **Q5:** Yes, the FLAME model we used includes teeth geometry. Each 3D Gaussian is bound to a triangle of the FLAME mesh, following GaussianAvatars (Line 123-125).
>
> ---
>
> Please let us know if you have any further questions.

---

> > ### Comment · Reviewer_mvUV · 2025-08-06
> > **Official Comment by Reviewer mvUV**
> >
> > Thank you for the authors’ response. While most of my concerns have been addressed, I would still like to emphasize the following point:
> >
> > 1. This work is largely built upon VASA-1. However, as VASA-1 is not open-sourced to date, this significantly limits the contribution and reproducibility of the current paper.

---

> ### Author Response · Authors · 2025-08-06
>
> Dear reviewer,
>
> Thank you again for your thoughtful feedback. As the discussion phase nears its end, we wanted to kindly bring your attention to our rebuttal and would be glad to provide any further clarification if needed.
>
> Authors

---

> ### Author Response · Authors · 2025-08-06
>
> Dear reviewer,
>
> Thank you for the additional feedback. We are glad to see that most of your concerns have been addressed! Regarding open-sourcing, as we mentioned in the rebuttal, we will offer our code and synthetic training data for research purposes. This will allow users to fully examine the details of our VASA-3D implementations and exactly reproduce our typical VASA-3D avatar samples and the driving sequences.
>
> In terms of contribution, we believe our core idea, which involves using a latent space learned from 2D videos to drive 3D talking faces, offers significant advantages over the prevailing approach of using 3DMMs. Our instantiation also contains important innovations to address the unique challenges we encountered, making our work the first of its kind to use a single real image and synthesized, imperfect videos to construct high-quality 3D talking faces. We believe that both our core idea and its implementation offer valuable insights that make a significant contribution to the community.
>
> Thank you again and please feel free to let us know if this response does not satisfactorily address your remaining concern or if there's any clarification we shall further provide that could assist with the final assessment. Thanks again.
>
> Authors

---

### Official Review · Reviewer_iSKP · 2025-07-02

**Clarity:** 3
**Significance:** 2
**Originality:** 2
**Rating:** 4
**Confidence:** 4

**Summary:**

This paper builds upon VASA-1 and 3D Gaussian Splatting (3DGS) to extend the task of talking head generation into the 3D domain. By leveraging VASA-1 to render a large number of video frames from a single input image, the authors train a 3DGS-based model. Furthermore, they utilize the motion latent from VASA-1 to generate realistic 3D talking head videos with relatively fast inference speed.

**Questions:**

1. Will the authors release the code and models introduced in this paper?
2. Since VASA-1 already supports some degree of head pose and camera movement control to produce pseudo-3D effects, how does the proposed method compare against VASA-1 under such settings?

**Ethical Concerns:**

["NO or VERY MINOR ethics concerns only"]

**Final Justification:**

I appreciate the authors' response. While the method has clear limitations, it demonstrates strong performance, and some of my initial concerns have been alleviated. Accordingly, I am inclined to moderately increase my score.

**Limitations:**

yes

**Quality:**

3

**Strengths And Weaknesses:**

**Strengths:**

1. The proposed method leverages VASA-1’s motion latent to enhance the expressiveness of facial motions in the 3DGS framework.
2. The paper presents several impressive demo videos that are visually convincing and contain relatively few artifacts under various head poses and expressions.
3. The proposed base deformation and VASA deformation modules are well-motivated and experimentally validated.

**Weaknesses:**

1. If I have not missed any key details, the proposed approach seems to heavily rely on VASA-1 to generate long video sequences from a single input image to train an avatar-specific 3D model. This significantly limits the method’s applicability, generalization ability, and efficiency in terms of storage and scalability.
2. While the use of motion latents from VASA-1 is claimed to be novel, it resembles prior approaches that condition 3DGS on dynamic face maps (e.g., SEGA[1], ExAvatar[2]) to compensate for the limited expressiveness of 3DMM-based representations. Additionally, the strong dependency on the non-open-source VASA-1 makes the method difficult to reproduce or benchmark.
3. Compared to some recent 3DGS-based avatar generation methods such as HRAvatar[3] and RGBAvatar[4], this method appears to require significantly more training video data.
4. From an implementation and benchmarking perspective, the paper overlooks a potentially fairer baseline: using an end-to-end diffusion-based talking head generation model[5,6]  to synthesize a training video from a single image, followed by training a 3DGS-based avatar model [3,4] from that video for comparison.
5. Although the authors introduce an SDS loss to mitigate artifacts from VASA-1 outputs (e.g., misalignments and large-pose distortions), this solution substantially increases the training cost. Moreover, Table 1 shows only marginal quantitative gains from this additional loss.

[1] SEGA: Drivable 3D Gaussian Head Avatar from a Single Image

[2] Expressive Whole-Body 3D Gaussian Avatar

[3] High-Quality and Relightable Gaussian Head Avatar

[4] RGBAvatar: Reduced Gaussian Blendshapes for Online Modeling of Head Avatars

[5] HunyuanPortrait: Implicit Condition Control for Enhanced Portrait Animation

[6] Hallo: Hierarchical Audio-Driven Visual Synthesis for Portrait Image Animation

---

> ### Author Rebuttal · Authors · 2025-07-29
>
> Thank you for the constructive feedback, and we address the questions as follows.
>
> **W1:** Thank you for the comment. We’d like to clarify that the video sequences used for model training do not necessarily need to be long. In fact, the model can be trained on only a collection of individual images (no videos) or a collection of short video clips (5-10 seconds long as done with random VoxCeleb2 samples in Sec. 4.1). Although we set the default training video length to 10 hours, it is shown in Fig. 3 that using only 2 hours leads to just a marginal performance drop, and we also used only 20-minute audio-generated continuous training video to compare with existing methods in Sec 4.3 (some of the compared methods require continuous video data due to their data preprocessing design and training loss implementation). Under these tested settings, the computational requirements are reduced significantly without much sacrifice in quality.
>
> We are unsure if this satisfactorily addresses your concern on reasons that may limit the "applicability, generalization ability, and efficiency". Please let us know if it doesn't or you have any further questions.
>
> **W2 & Q1:** We thank the reviewer for pointing out the related works SEGA [1] (a concurrent and arXiv work, April 2025) and ExAvatar [2], which we will cite in our revised paper. However, with due respect, we find significant differences between these works and ours, in terms of their motivations and solutions to enhance the expressiveness of 3D parametric models.
>
> The key motivation of our work is that the parameter space of existing 3D parametric models is intrinsically limited and cannot capture nuanced motion and subtle expressions of real human faces when talking (See Line 4-5, Line 25-33, Line 40-42). So we advocate *the use of the powerful latent space learned on massive 2D talking face videos to drive 3D Gaussian avatars*. To the best of our knowledge, this is a novel technical contribution not explored by previous works, and we have demonstrated its clear advantage (Table 1, Fig. 5, etc.).
>
> Please note that both [1] and [2] still rely on FLAME's expression parameters to drive their 3D models. Even though their dynamic offsets are learned from the subject’s data, their raw driving signals still stem from the FLAME expression space. These offsets lead to results that more closely adhere to the subject’s appearance, but they do little to enhance the expression space to encompass subtle expressions and nuanced motions. In contrast, our 3D model is driven by learned latent codes (from VASA-1), not FLAME expression codes.
>
> Regarding open-sourcing, we plan to release the code and some synthetic training data of VASA-3D for research purposes, which will facilitate checking the details of our VASA-3D implementations and exactly reproducing some typical VASA-3D avatars and the driving sequences shown in our paper.
>
> **W3:** While more training data was used in our method (minimum 20-minute videos containing 30K frames tested our paper) than HRAvatar [3] or RGBAvatar [4] (typically 2K-5K frames), there are key differences in task setup and data assumptions. These two methods use carefully captured monocular videos with exaggerated expressions and diverse poses to improve data coverage. Moreover, their training and testing data often come from highly similar distributions, enabling efficient data use.
>
> In contrast, we simply take random real-world talking videos or audios to generate our training data, which represent an easier and more scalable data capture setup than these methods. We also use random real-world audios for testing. Under such situation, it is natural to use more training data to cover a wide motion distribution including nuanced facial expressions.
>
> Additionally, our method must handle the inconsistencies in VASA-1-generated synthetic videos, further increasing the complexity. This makes our setting different and more challenging than these 3DGS avatar works.
>
> **W4:** We thank the reviewer for suggesting this potential baseline approach. We are unable to add this baseline in a short time window. However, we wish to make a few arguments:
>
> - Once again, our key motivation is to build expressive 3D head avatars animated with powerful latent spaces beyond 3DMM parameters. We have demonstrated the limitations of 3DMM parameters in expressing nuanced motions (Table 1, Fig. 5, etc.). As far as we can see, both [3,4] still use FLAME parameters to drive 3D facial expression, similar to the base model we used, i.e., GaussianAvatars. In fact, we view [3,4] as technical advancements orthogonal to ours and they can be applied as our base model in lieu of GaussianAvatars.
>
> - Using a large, diffusion-based talking head generation model such as [5,6] to generate data will incur significantly increased computation cost compared to using VASA-1 (e.g., <1 fps generation using [6]). Moreover, using such data still does not resolve the aforementioned challenge.
>
> - It's not clear to us how to adapt such baselines to the *audio-driven* scenarios. The task we handle in this work is audio-driven 3D talking face generation, with the target application being virtual, autonomous conversational agents. We provide more discussions related to audio-driven and video-driven methods at the end of this response.
>
> **W5:** The main goal of the SDS loss is to mitigate visual artifacts in side views (Line 189). Since the enhanced regions only occupy small image areas in side views, their improvement is not well reflected by the numerical values computed on all images with both frontal and side views. However, the visual improvements can be clearly observed (Fig. 4). We will add explanations in the revision, and we thank the reviewer for this comment.
>
> **Q1:** Please see above.
>
> **Q2:** There are multiple advantages of VASA-3D compared to VASA-1 in terms of 3D-related capabilities:
> - There's no true camera viewpoint control in VASA-1. The head pose control therein primarily steers the head angle with shoulder and torso regions largely unchanged. The camera distance control was approximated with 2D head region size as conditioning signal. In contrast, VASA-3D supports true 3D camera control and free-view rendering.
>
> - VASA-3D can be naturally integrated into 3D environment (VR/AR/Games) as it's inherently a 3D model with 3DGS rendering.
>
> - Last but not least, even when rendered as 2D videos, VASA-3D's results exhibit strong texture consistency. In contrast, VASA-1 shows noticeable texture sticking artifacts under head rotations and large facial motions, especially around regions with high-frequency textures such as hair and beard.
>
> We will add more related discussions in our revised paper, and we thank the reviewer for raising this question.
>
> ---
>
> **Motivation, Contribution, and Relationship to Video-Driven Methods:**
>
> Here we'd like to rephrase the motivation and main contribution of our paper. First, we observed that existing *audio-driven* 3D talking head methods cannot achieve highly realistic and vivid talking head results. The quality is far worse than the 2D talking face counterpart. After examination, we found that most existing methods apply 3D parametric models, in particular, FLAME, for expression control, following a generic "audio->FLAME expression->deformed 3D model" paradigm. However, existing 3D parametric models like FLAME are constructed by registering 3D scans – typically from just a few hundred subjects and without sophisticated expression design – and then applying PCA. They have intrinsic limitations in handling nuanced motion and subtle expressions of real human faces when talking and expressing emotions. So we propose to use a latent space learned on massive 2D face videos to drive 3D avatars, an attempt not made by any previous methods. We also propose a framework to map the learned latents to 3D Gaussian variations (Sec. 3.2) and a set of carefully designed losses to handle the training on synthesized data and addressing the challenges they incur (Line 170).
>
> We have demonstrated the clear advantage of our method compared to using FLAME parameters. We also achieved high-quality and naturalistic audio-driven 3D talking head animation results from a single image, significantly pushing the state-of-the-art quality in this domain and minimizing the performance gap to audio-driven 2D talking head methods.
>
> We also would like to re-emphasize that the task we handle in this work is *audio-driven talking face* generation. We acknowledge that there is rapid progress in 3D avatar reconstruction and video-based reenactment, such as those works pointed out by the reviewer. However, please note that audio-driven and video-driven 3D head avatar generation are often treated as distinct problems. As we mentioned earlier, those works can potentially be applied as our base models. In fact, we believe that *adapting these works to the audio-driven scenario is not trivial and achieving high realism and vividness in the talking scenario introduces challenges not fully addressed by these works*. Taking our base model GaussianAvatars as an example,  their video results on expression transfer (with FLAME expression codes) look convincing (see demos on their website). However, when it comes to *audio-synchronized motion behaviors and the expression of facial nuances subtle yet critical for expressing emotions*, FLAME's expression space is insufficient, as shown in our paper.
>
> We are open to any discussion and please let us know if you have any further comments.

---

> > ### Comment · Reviewer_iSKP · 2025-08-06
> >
> > After reviewing the comments from other reviewers, I acknowledge that the current method indeed has several limitations:
> >
> > 1. The approach heavily depends on VASA-1, which is not open-sourced.
> >
> > 2. It requires per-subject training, limiting its generalizability.
> >
> > Although some reviewers gave positive scores, their comments are rather brief and lack substantive insight.
> >
> > That said, the effectiveness of the method is now widely recognized, and the authors have committed to releasing the code to facilitate future research. I am willing to trust the authors on this point. As such, I believe most of my initial concerns have been adequately addressed.
> >
> > Overall, I am inclined to raise my score accordingly.

---

> > > ### Author Response · Authors · 2025-08-06
> > >
> > > Dear reviewer,
> > >
> > > Thank you for your feedback on our rebuttal. We are glad to know that most of your initial concerns have been adequately addressed! Regarding "per-subject training", we acknowledge that it introduces limitations on broad applicability due to the training stage. Still, we wish to point out that existing audio-driven 3D talking avatars mostly requires per-subject training (e.g., the audio-driven methods we compared in the main paper including ER-NeRF, GeneFace, MimicTalk, and TalkingGaussian). We will explore solutions not requiring per-subject training in our future work, such as applying a meta-network which uses the input image to control the parameters of the 3D Gaussian avatar's MLP networks in a feed-forward manner.
> > >
> > > Thank you again for all the insightful comments and suggestions! We will revise our paper accordingly, incorporating our clarifications and the additional references mentioned by the reviewer.
> > >
> > > Authors

---

> ### Author Response · Authors · 2025-08-06
>
> Dear reviewer,
>
> Thank you again for your thoughtful feedback. As the discussion phase nears its end, we wanted to kindly bring your attention to our rebuttal and would be glad to provide any further clarification if needed.
>
> Authors

---

### Official Review · Reviewer_trR7 · 2025-07-02

**Clarity:** 4
**Significance:** 3
**Originality:** 3
**Rating:** 5
**Confidence:** 4

**Summary:**

This paper presents a method to create a human head avatar from a single image that can be driven by audio. Given an input image, it uses a 2D diffusion model, VASA-1, to generate video frames to form the training dataset, which is then used to optimize a 3DMM-driven 3DGS avatar model, GaussianAvatars. With the aid of the powerful motion space of VASA-1, this method can capture a wider range of facial expressions and body movements, which better align with the audio input. The authors also explore several regularization techniques to enhance visual details without harming 3D consistency. Solid experiments and sufficient visual results are provided to exhibit the effectiveness of this method.

**Questions:**

- Given the existence of two separate networks for the face and the neck, how can one tell if a Gaussian splat belongs to the face or the neck?

**Ethical Concerns:**

["NO or VERY MINOR ethics concerns only"]

**Final Justification:**

3DGS avatars often take many views and frames to become robust to viewing points and animations, while simply plugging in the SDS loss leads to artifacts in unseen views. This method combines the generative ability of VASA-1 with the 3D consistency of 3DGS, producing impressive visual and quantitative results. Given that the concerns of myself and other reviewers are resolved, I am glad to still vote for acceptance.

**Limitations:**

yes

**Quality:**

4

**Strengths And Weaknesses:**

# Strengths
1. This is a well-written paper with a strong motivation and impressive execution. The visual results, especially the video demos, are very convincing - the rendered avatars show great 3D consistency, sharp details, and expressive facial movements that align well with the audio.
1. The method is concise and self-contained - each component appears for a clear reason and is consolidated with sufficient experiments. The long list of loss terms seems verbose, but they are actually all necessary, as shown in Table 1 and Figure 4&5.
1. It is very nice to see the rendering of "Base Deform" and "+VAS Deform" side-by-side in Figure 4, which clearly shows the advantage of using the broader motion space of VASA-1.
1. The ablation study of dataset size and number of iterations shown in Figure 3 is a good example of scaling up in an effective way.

# Weaknesses
1. The efficiency and  latency of this method during inference is not reported. Does it support realtime audio-driven synthesis given a trained avatar?

---

> ### Author Rebuttal · Authors · 2025-07-29
>
> Thank you for the positive feedback on our work. We appreciate your recognition of the contribution and effectiveness of our method. We address your questions as follows.
>
> **W1:** Given an audio clip and a trained VASA-3D model, our method can generate rendered frames at **75 fps** (i.e., ~13ms for one frame) with a **65ms** latency. The speed and latency were measured on one RTX 4090 GPU; see Line 282 of the main paper. So yes, **it supports real-time audio-driven synthesis**. A recorded real-time demo is also provided in the supplementary video ("1.Realtime_Demo.mp4").
>
>
> **Q1:** The face and neck regions are pre-defined on the FLAME template mesh. Since 3D Gaussians are bound to the triangles of the deformed FLAME mesh, the regions they belong to can be easily derived.
>
> ---
>
> Kindly let us know if you have any further questions.

---

> > ### Comment · Reviewer_trR7 · 2025-08-08
> >
> > I thank the authors for the clarification. My questions about the efficiency and region assignment strategy have been well resolved.

---

### Official Review · Reviewer_9EUc · 2025-07-13

**Clarity:** 4
**Significance:** 4
**Originality:** 4
**Rating:** 5
**Confidence:** 3

**Summary:**

The paper introduces VASA-3D, a method for generating lifelike, audio-driven 3D head avatars from a single portrait image. The key innovation lies in leveraging the motion latent space of VASA-1 to animate a 3D Gaussian splatting model. By synthesizing diverse facial expressions and poses via VASA-1, the method trains a 3D head model that supports free-viewpoint rendering and real-time animation. A robust training framework with tailored losses addresses challenges like temporal inconsistency and limited pose coverage in synthetic data. Experiments demonstrate superior realism and performance compared to existing methods, with 512×512 videos rendered at 75 FPS.

**Questions:**

1. How does VASA-3D handle identity preservation when synthesizing expressions not present in the input image?
2. What are the trade-offs between using FLAME as a base model and alternative parametric representations?
3. Can the method generalize to non-frontal input images or low-resolution portraits?
4. How sensitive is the training to the choice of VASA-1 as the motion latent source?

**Ethical Concerns:**

["NO or VERY MINOR ethics concerns only"]

**Final Justification:**

I appreciate the authors' detailed response. Most of my concerns got addressed fairly. I keep the final score.

**Limitations:**

1. Synthetic Data Dependency: Relies on VASA-1 for training data, which may not capture all real-world variations.
2. High Resource Requirements: Time and computational demands for training may restrict accessibility.
3. Artistic Image Support: While tested on artistic portraits, real-world applicability remains unexplored.

**Quality:**

4

**Strengths And Weaknesses:**

Strengths:
1. Single-Image Customization: Achieves high-quality 3D head avatars from a single image, overcoming the need for multiview data.
2. Integration of 2D and 3D: Effectively translates 2D motion latent into 3D deformations, enabling expressive animations.
3. Real-Time Performance: Supports free-viewpoint rendering at 75 FPS, making it practical for immersive applications.
4. Strong Empirical Results: Outperforms state-of-the-art methods in metrics like FID, lip-sync accuracy, and user studies.

Weaknesses:
1. Reliance on VASA-1: Synthetic data quality depends on VASA-1, which may inherit its limitations.
2. Computational Cost: Requires extensive synthetic data (10 hours per portrait) and 18-hour training, limiting scalability.
3. Limited Generalization: Tested primarily on StyleGAN2-generated portraits; performance on real-world images is unverified.
4. Code Unavailability: VASA-1 is not open sourced

---

> ### Author Rebuttal · Authors · 2025-07-29
>
> Thanks for your recognition of our work. We appreciate your positive feedback and address your questions as follows.
>
>
> **W1 & L1:** Thank you for the comment. Indeed, using synthetic data created by VASA-1 (or other talking head video generation method) may introduce distribution gaps to real data. We'll add more related discussions in the limitation section. Nevertheless, we found that our training data enables VASA-3D to generate vivid and realistic talking heads that exhibit clear superiority over other techniques. Also note that VASA-3D could also leverage future advances in 2D talking head generation for improved training.
>
> **W2 & L2:**
> Since training is done offline, efficiency was not our main focus. However, we show in the following that strong results can still be achieved with much less computation.
>
> Training data size and generation speed:
>
> Although 10 hours of training video is the default setting, it is shown in Fig. 3 that similar results could be obtained with just 2 or 5 hours of data. Note that training data generation is fast: <1 hour for 10 hours of video (Line 244), or ~10 min for 2 hours of data.
>
> Furthermore, our comparisons with other methods in Sec. 4.3 used only *20 minutes* of audio-generated video data for training (because some of the methods require continuous video and cannot utilize the collection of video clips in our 10-hour data). The results show that VASA-3D trained with 20-min data still generates highly realistic 3D talking heads, surpassing the other methods by a wide margin across metrics evaluating lip-sync, video quality, etc. (see "5. Methd_Comparisons.mp4" and Table 3). The 20-min data can be generated in <2min.
>
> Training time:
>
> We wish to point out that 18 hours of training time is reasonable for 3D model optimization on videos. For reference, we report the training time of different methods in Table 3 of the main paper. For the compared methods, we stop the training when convergence was observed on our training data. Note that we kept their original implementation of running on a single GPU unchanged.
>
> We did not carefully optimize the training framework or tune the hyperparameters for efficiency considerations. But the training time can be reduced with more GPUs or implementation optimizations (e.g., less frequent SDS loss computation which is the most costly part).  For the comparison below, we found that our model trained with 20K iterations (1.8 hours) already significantly outperforms other methods and nearly reaches the performance of the model trained with 200K iterations.
>
> | |$S_C\uparrow$  |$S_D\downarrow$ | FID $\downarrow$ |
> | ---- |---- |---- |---- |
> |  ER-NeRF (26h, 1 GPU) |  5.921 |8.7788 | 18.36 |
> |  GeneFace (25h, 1 GPU) |  5.922 | 9.6066 | 26.33 |
> |  MimicTalk (32h, 1 GPU) | 5.270 |10.9368 | 20.85 |
> |  TalkingGaussian (3h, 1 GPU) |  6.701 | 8.1061 | 23.62 |
> |  Ours-20K (1.8h, 4GPU)  | 7.980 |7.0020 | 10.35
> |  Ours-200K (18h, 4GPU) |  8.121 | 6.9300 | 7.45 |
>
> In summary, the training cost is reasonable, and significant computational reductions (down to 20 minutes of training video and 1.8 hours of training) still lead to high quality results. We intend to study efficiency improvements in future work.
>
> **W3 & L3:** There were experiments on real-world images, and the performance was equally good to StyleGAN2-generated portraits. Please see Line 36, Table I, and Fig. V in the supplementary pdf, and the video results in "2.Main_Results.mp4" (02:03-02:59).
>
> **W4**: We plan to release the code and some processed synthetic training data of VASA-3D for research purposes, which will facilitate checking the details of our VASA-3D implementations and exactly reproducing some typical VASA-3D avatars and the driving sequences shown in our paper.
>
> **Q1:** Identity preservation under novel expressions is primarily handled by VASA-1. The synthetic training videos generated by VASA-1 effectively preserve identity with novel head poses and expressions, as demonstrated in their paper.
>
> **Q2:** FLAME is a well-established 3D morphable model that provides a parametric representation of human faces, and it's widely used in previous 3D Gaussian avatar methods such as [a,b,c]. We followed these works to use FLAME and did not try alternative parametric models. Other parametric models are applicable as well and we'll explore the trade-offs in our future work.
>
>
> **Q3:** Yes, VASA-3D can generalize to non-frontal input images (with yaw angles up to ~45 degrees) and low-resolution portraits.
>
>
> **Q4:** The choice of VASA-1 as the motion latent source is based on its proven ability to generate expressive talking videos with diverse head poses and expressions, which are essential for supervising our 3D Gaussian model training. Although we have not explicitly tested other motion latent sources, our method does not depend on any specific architecture of VASA-1 or any specific properties of VASA-1's latent space.
>
>
> [a] S Qian et al. GaussianAvatars: Photorealistic Head Avatars with Rigged 3D Gaussians
>
> [b] PW Grassal et al. Neural Head Avatars from Monocular RGB Videos
>
> [c] Y Zheng et al. I M Avatar: Implicit Morphable Head Avatars from Videos
>
> ---
>
> Thank you once again and please let us know if you have any further questions.

---

### Decision · Program_Chairs · 2025-09-17

**Decision:**

Accept (poster)

**Comment:**

(a, b) Precise alignment of audio and visuals in realistic avatars is an unsolved problem, although there is progress. This work approaches optimizing the rendering in VASA latent space with audio/speech driven physics based guidance (via gaussian splat smoothing). This conditioning of video gen on audio is important finding. The explicit parameterization of facial expressions in the splats is also key. The paper is written well; and the video demos are compelling (clearly the authors put in rigor into the work).

(c) There are some related works and baseline that were missing; as were pointed out by the reviewers. Authors addressed these in the rebuttal. Its not clear how real-time and generalizable this can be in scenarios where audio/speech inputs are noisy (a potential next work, this does not diminish this paper's contributions). The work is directly building on VASA-1, and so, that also limits the overall generalization capacity.

(d) Impact in single image/view driven synthesis of long form video. Shows ways to address realism of facial expressions in Audio visual context generation. Beating state of the art. Clarity of presentation and rigor in evaluations.

(e) Bulk of the reviewer questions were re: missing related work, claims about novelty (which were addressed), corrections in paper presentations, and 1-2 potential missing baselines (which do not change the overall story arc). These all help clarify the work but they are mostly procedural. No technical flaws are highlighted.